# 12-*O*-tetradecanoylphorbol-13-acetate Reduces Activation of Hepatic Stellate Cells by Inhibiting the Hippo Pathway Transcriptional Coactivator YAP

**DOI:** 10.3390/cells12010091

**Published:** 2022-12-26

**Authors:** Chang Wan Kim, Yongdae Yoon, Moon Young Kim, Soon Koo Baik, Hoon Ryu, Il Hwan Park, Young Woo Eom

**Affiliations:** 1Department of Thoracic and Cardiovascular Surgery, Yonsei University Wonju College of Medicine, Wonju 26426, Republic of Korea; 2Regeneration Medicine Research Center, Yonsei University Wonju College of Medicine, Wonju 26426, Republic of Korea; 3Department of Internal Medicine, Yonsei University Wonju College of Medicine, Wonju 26426, Republic of Korea; 4Department of Surgery, Yonsei University Wonju College of Medicine, Wonju 26426, Republic of Korea

**Keywords:** 12-*O*-tetradecanoylphorbol-13-acetate, hepatic stellate cell, protein kinase Cδ, Yes-associated protein 1

## Abstract

Although protein kinase C (PKC) regulates various biological activities, including cell proliferation, differentiation, migration, tissue remodeling, gene expression, and cell death, the antifibrotic effect of PKC in myofibroblasts is not fully understood. We investigated whether 12-*O*-tetradecanoylphorbol-13-acetate (TPA), a PKC activator, reduced the activation of hepatic stellate cells (HSCs) and explored the involvement of the Hippo pathway transcriptional coactivator YAP. We analyzed the effect of TPA on the proliferation and expression of α-smooth muscle actin (SMA) in the LX-2 HSC line. We also analyzed the phosphorylation of the Hippo pathway molecules YAP and LATS1 and investigated YAP nuclear translocation. We examined whether Gö 6983, a pan-PKC inhibitor, restored the TPA-inhibited activities of HSCs. Administration of TPA decreased the growth rate of LX-2 cells and inhibited the expression of α-SMA and collagen type I alpha 1 (COL1A1). In addition, TPA induced phosphorylation of PKCδ, LATS1, and YAP and inhibited the nuclear translocation of YAP compared with the control. These TPA-induced phenomena were mostly ameliorated by Gö 6983. Our results indicate that PKCδ exerts an antifibrotic effect by inhibiting the Hippo pathway in HSCs. Therefore, PKCδ and YAP can be used as therapeutic targets for the treatment of fibrotic diseases.

## 1. Introduction

Hepatic stellate cells (HSCs), which are liver resident fibroblasts, are located in the space of Disse between hepatocytes and sinusoidal endothelial cells and constitute 5–8% of total liver resident cells [1,2]. Quiescent HSCs in the normal liver play an important role in supporting liver development and regeneration, vitamin A storage, intracellular lipid droplet storage, immunoregulation, and liver hemodynamic homeostasis [3,4]. Liver injury stimulates the transdifferentiation of quiescent into activated HSCs. During activation, the properties of HSCs change, including their proliferation, ECM production, migration towards chemokines, contraction, and loss of lipid droplets, ultimately promoting hepatic fibrosis [5,6,7,8,9,10]. As the quantity of activated HSCs is known to correlate with fibrosis severity [11,12], controlling the activity of HSCs is considered essential for the treatment of liver fibrosis.

Protein kinase C (PKC) is a group of serine/threonine protein kinases that play important roles in various biological functions, including cell proliferation, differentiation, migration, gene transcription and translation, and apoptosis [13,14,15,16]. PKC isoforms are grouped into four classes: conventional (cPKCs; α, βI, βII, and γ), novel (nPKCs; δ, ε, η, and θ), atypical (aPKCs; ζ, λ/ι), and PKCμ (a form between novel and atypical isoforms) isoforms [17]. Of note, cPKCs are Ca^2+^- and diacylglycerol-dependent, nPKCs are Ca^2+^-independent and diacylglycerol-dependent, whereas aPKCs are Ca^2+^- and diacylglycerol-independent. The expression of PKC isoforms is known to differ not only between species, but also between normal and disease states in the same organism [18]. Most studies have suggested that PKCs are associated with the malignant transformation of cells, including breast [19], lung, and gastric carcinomas [20], as well as with the promotion of fibrosis [21,22,23].

The Yes-associated protein 1 (YAP)/transcriptional coactivator with PDZ-binding motif (TAZ) signaling has been linked to the pathophysiology of fibrosis. Interestingly, aberrant activation of YAP/TAZ has been observed in human fibrotic tissues (i.e., lung, kidney, skin, and liver), animal models of fibrosis (i.e., bleomycin-induced lung or skin fibrosis, diabetic nephropathy, and nonalcoholic steatohepatitis), and various cultured cells (i.e., fibroblasts, epithelial cells, keratinocytes, and hepatocytes) [24,25,26,27,28,29,30,31,32,33,34,35]. Activation of YAP, which is represented by its translocation to the nucleus, is a critical driver of hepatic stellate cell activation, leading to α-SMA expression [36,37]. The phosphorylation of YAP at Ser-127 promotes its cytoplasmic retention, whereas phosphorylation at Ser-397 induces its degradation [38].

12-*O*-tetradecanoylphorbol 13-acetate (TPA) is used as a cPKC or nPKC activator because it can substitute diacylglycerol by directly binding to the C1 domain, leading to PKC activation. TPA is widely used as an immune cell stimulator and for understanding the cell-specific function of PKC isoforms. TPA can increase the cell proliferation of malignant cells in several types of tumors, and thus, acts as a promotor for melanoma, breast cancer, oral cancer, and skin carcinogenesis [39,40,41]. In contrast, treatment with TPA decreases the proliferation capacity of lymphoma and liver cancer cells [42,43,44]. Many studies have reported that PKCs promote fibrosis [45,46,47]; however, TPA was recently reported to weaken the proliferation and transdifferentiation of cardiac fibroblasts through cPKC and nPKC [48]. In addition, we found that the proliferation of LX-2 hepatic stellate cells and expression of α-SMA were both suppressed when LX-2 cells were cocultured with TPA-induced macrophages from monocytic THP-1 cells [49]. 

Based on these findings, we hypothesized that PKC activity could alleviate liver fibrosis. Hence, we examined whether induction of PKC activity in TPA-treated LX-2 hepatic stellate cells alleviates fibrosis in vitro. To this end, we investigated whether TPA suppresses the proliferation of LX-2 cells and expression of α-SMA and whether the Hippo pathway transcriptional coactivators YAP/TAZ are involved in the regulation of this activity. Through comprehensive analysis, we aimed to evaluate whether the induction of PKC activity can be used as a therapeutic approach for the treatment of liver fibrosis.

## 2. Materials and Methods

### 2.1. Materials

TPA and methylthiazolyldiphenyl tetrazolium bromide (MTT) were obtained from Sigma-Aldrich (St. Louis, MO, USA), while TGF-β was purchased from R&D Systems (Minneapolis, MN, USA). Antibodies against α-SMA (ab7817) were from Abcam (Cambridge, UK), antibodies against PKCδ (sc396), YAP (sc101199), Lats1 (sc398560), and GAPDH (sc47724) were from Santa Cruz Biotechnology (Santa Cruz, CA, USA), while antibodies against PKCs (phospho-PKC antibody sampler kit, 9921T), COL1A1 (39952S), pLats1 (8654S), pYAP(4911S), and horseradish peroxidase-conjugated secondary antibodies (7074S and 7076S) were obtained from Cell Signaling Technology (Danvers, MA, USA). ProLong™ Gold Antifade Mountant with DAPI (4’,6-diamidino-2-phenylindole; P36930) was purchased from Invitrogen (Waltham, MA, USA), while the pan-PKC inhibitor Gö 6983 (HY-13689) was acquired from MedChemExpress (Princeton, NJ, USA). All other materials were purchased from Sigma-Aldrich (St. Louis, MO, USA) unless otherwise indicated.

### 2.2. Cell Culture

The LX-2 human HSC line was purchased from Millipore (Burlington, MA, USA). LX-2 cells were maintained in Dulbecco’s modified Eagle medium (DMEM, Gibco BRL, Rockville, MD, USA) supplemented with 3% fetal bovine serum (FBS, Gibco BRL) and penicillin/streptomycin (Gibco BRL) at 37 °C and 5% CO_2_. For experiments, LX-2 cells were seeded for 24 h, and treated with TPA for an additional 48 h. Cells were exposed to Gö 6983 20 min prior to TPA treatment. 

### 2.3. MTT Assay

Cells were seeded at a density of 1 × 10^4^ cells/cm^2^ in 96-well plates and treated with Gö 6983 to inhibit the activation of PKCs, as mentioned in Section 2.2. MTT dissolved in phosphate-buffered saline (PBS) was added to each well (final concentration: 5 mg/mL), and cells were further incubated at 37 °C for 2 h. MTT formazan was dissolved in 100 μL dimethyl sulfoxide following incubation for an additional 15 min with shaking. Subsequently, the optical density of each well at 570 nm was measured using a microplate reader (Molecular Devices; San Jose, CA, USA). 

### 2.4. qPCR

Total RNA was extracted using TRIzol reagent (Gibco BRL) according to the manufacturer’s instructions. cDNA was synthesized from 1 μg total RNA using the Verso cDNA synthesis kit (Thermo Fisher Scientific, Waltham, MA, USA). The expression of α-*SMA*, *COL1A1*, *COL3A1*, and *GAPDH* was evaluated using the respective sense and antisense primers (Table 1). The reaction mixture (10 μL) included cDNA, primer pairs, and SYBR Green PCR Master Mix (Applied Biosystems, Dublin, Ireland), and PCR was conducted using a QuantStudio 6 Flex Real-time PCR System (Thermo Fisher Scientific). All qPCR reactions were performed in triplicates. The expression of *GAPDH* was used for normalization. The 2^−(ΔΔCq)^ method was used to calculate the relative fold changes in mRNA expression.

### 2.5. Immunoblotting Analysis

Cells were lysed in sample buffer [62.5 mM Tris-HCl, pH 6.8, 34.7 mM sodium dodecyl sulfate (SDS), 10% (*v*/*v*) glycerol, and 5% (*v*/*v*) β-mercaptoethanol], boiled for 5 min, subjected to SDS-polyacrylamide gel electrophoresis, and transferred to an Immobilon membrane (Millipore). After blocking with 5% skim milk in Tris-HCl-buffered saline containing 0.05% (*v*/*v*) Tween 20 (TBST) for 30 min, the membrane was incubated with primary antibodies against α-SMA, Lats1, PKCδ, pYAP, YAP, and GAPDH at a dilution of 1:1000 or COL1A1, pLats1, pPKCδ, pPKCα/β, pPKCθ, and pPKCζ/λ at a dilution of 1:2000 at 4 °C overnight. The membrane was washed thrice for 5 min with TBST and then incubated with horseradish peroxidase-conjugated secondary antibodies (1:5000) for 1 h. After washing thrice with TBST, protein bands were visualized using an EZ-Western Lumi Pico or Femto kit (Dogen, Seoul, Republic of Korea) according to the manufacturer’s instructions and detected using a ChemiDoc XRS+ system (Bio-Rad, Hercules, CA, USA). 

### 2.6. Immunocytochemical Analysis

To detect the localization of the YAP protein, LX-2 cells were grown on glass coverslips and exposed to TPA with or without Gö 6983 for 24 h. Cells were then washed with PBS, fixed with 4% paraformaldehyde, permeabilized with 0.2% Triton X-100 for 10 min at 25 °C, and blocked with 3% FBS in PBS for 30 min, and then incubated with a primary antibody specific for YAP (1:100; Santa Cruz) at 4 °C overnight. For fluorescence labeling, cells were incubated with Alexa Fluor 488-conjugated secondary antibody (1:100; Invitrogen, Carlsbad, CA, USA) for 1 h at 25 °C. The unbound secondary antibody was removed by washing, and cells were mounted in ProLong™ Gold Antifade Mountant with DAPI. Cells were observed and photographed under a fluorescent microscope (Eclipse TS2R, Nikon, Tokyo, Japan).

### 2.7. Cell Cycle Analysis

Cellular DNA contents were analyzed using the CycleTEST plus DNA reagent kit (BD Biosciences, San Jose, CA, USA) according to the manufacturer’s instructions. Briefly, LX-2 cells were trypsinized and centrifuged at 2000 rpm for 5 min. Cells were washed twice with the buffer solution provided in the kit and sequentially treated with solutions A, B, and C according to the manufacturer’s instructions. DNA contents were analyzed on a flow cytometer (BD FACSAria III, BD Biosciences). 

### 2.8. Statistical Analyses

All experiments were performed thrice. Data are expressed as the mean ± standard deviation. *p*-values were determined using a paired, two-tailed Student’s *t*-test or a Mann–Whitney U test. All statistical analyses were performed using the GraphPad Prism 7.0 software (GraphPad Inc., La Jolla, CA, USA). Significance was set at *p* ≤ 0.05.

## 3. Results

### 3.1. TPA-Induced Inhibition of Activation of Hepatic Stellate Cells

Although TPA promotes tumor progression and fibrosis, the opposite effects can also occur depending on cell type [44,48]. As a preliminary study, we analyzed the effect of TPA on the proliferation of two hepatocellular carcinoma cell lines (HepG2 and Huh-7), hepatic stellate cells (LX-2), and cardiac fibroblasts. When we treated each cell line with 10 nM TPA for 2 days, we found that the proliferation of HepG2 was increased by approximately 36%, whereas that of Huh-7 cells was not affected. Conversely, treatment of LX-2 cells and cardiac fibroblasts with TPA resulted in a decrease in their proliferation by 27% and 39%, respectively (Appendix A). Based on these results, we investigated the mechanisms of TPA on the proliferation of LX-2 cells and expression of α-SMA. We detected that TPA at concentrations above 1 nM decreased the proliferation of LX-2 cells (Figure 1a), with 10 nM TPA retarding the proliferation rate of LX-2 cells compared with that of the control group in a time-dependent manner (Figure 1b). When we used flow cytometry to analyze the cell cycle at the indicated timepoints after TPA treatment, we found that the S and G2/M phases were increased by approximately 3.2% and 6.3% at 12 h, whereas the G1 phase was decreased by approximately 10.1%. Of note, the cell cycle was restored to levels similar to those of the control group at 48 h. Since TPA decreased the proliferation rate of LX-2 cells (Figure 1a,b) and we rarely observed the apoptotic population in TPA-treated LX-2 cells (Figure 1c), the increase in the G2/M phase at 12 h suggested that TPA was transiently associated with G2/M arrest in LX-2 cells. In addition, we observed that TPA decreased the expression of *α-SMA*, *COL1A1*, and *COL3A1* mRNA (Figure 1d), and in particular inhibited the expression of α-SMA and COL1A1 in a dose-dependent manner (Figure 1e). These results suggest that TPA induces transient G2/M arrest in LX-2 cells, slowing cell proliferation and effectively inhibiting the expression of α-SMA.

### 3.2. TPA-Induced Phosphorylation of PKCδ and YAP in HSCs

TPA acts as a PKC activator [50] and PKCs inhibit the activity of YAP/TAZ [51], which have been associated with the pathophysiology of fibrosis [24,52,53]. Therefore, we investigated which PKC isoforms were activated and whether the phosphorylation or localization of YAP was changed by TPA in LX-2 cells. To detect the phosphorylation of PKCs, we treated LX-2 cells with 10 nM TPA and lysed them at the indicated timepoints. We found that the phosphorylation of PKCα/β, PKCθ, and PKCζ/λ was gradually decreased with time compared with that in the control group, whereas the phosphorylation of PKCδ was increased from baseline at 30 min to maximal phosphorylation at approximately 12 h (Figure 2). Similar to the phosphorylation of PKCδ, we detected that the total levels of PKCδ gradually increased over 12 h. (Figure 2). These results suggest that TPA regulates the activity of LX-2 cells through the regulation of the activity of PKCδ. As YAP acts downstream of PKCs and the peak phosphorylation of PKCδ was observed at 12 h, we analyzed both the phosphorylation and intracellular localization of YAP on day 1 after TPA treatment. We detected the phosphorylation of LATS and YAP when cells were treated at concentrations above 1 nM TPA (Figure 3a). We observed a significant portion of YAP in the nuclei of cells in the control group, whereas the fluorescence brightness of YAP in the nuclei of TPA-treated LX-2 cells was decreased compared with that in the control group (Figure 3b). These results suggest that greater than 1 nM of TPA increases YAP phosphorylation, which promotes its hydrolysis and further inhibits its nuclear localization, thereby suppressing gene expression, including that of α-SMA.

### 3.3. Roles of PKCδ and YAP in HSC Activation

Next, we investigated whether the pan-PKC inhibitor, Gö 6983, could restore the TPA-induced inhibition of proliferation and decrease in the expression of α-SMA in LX-2 cells. To detect whether Gö 6983 inhibited the phosphorylation of PKCδ, we exposed LX-2 cells to 1 μM Gö 6983 20 min prior to TPA treatment. We found that the TPA-induced phosphorylation of PKCδ was abolished by Gö 6983 at 12 h (Figure 4c). In addition, the TPA-induced reduction in cell proliferation was recovered to control levels by Gö 6983 (Figure 4a), while also no G2/M arrest was observed in LX-2 cells treated with TPA and Gö 6983 at 12 h (Figure 4b). We further noticed that the decrease in the expression of *α-SMA*, *COL1A1*, and *COL3A1* mRNA as well as the levels of α-SMA and COL1A1 proteins were also restored by Gö 6983 (Figure 4d,e). These results suggest that the activation of PKCδ plays an important role in the regulation of the activation of LX-2 cells, which is represented by an increase in proliferation potential and expression of α-SMA.

Furthermore, it was observed that Gö 6983 decreased the TPA-induced phosphorylation of YAP (Figure 5a), with most YAP being located in the nuclei of LX-2 cells, similar to the control group (Figure 5b). These results suggest that YAP phosphorylation inhibits the translocation of YAP into the nucleus. Therefore, TPA induces phosphorylation of PKCδ and YAP to inhibit the translocation of YAP into the nucleus, leading to growth suppression and a decrease in the expression of α-SMA in LX-2 cells. Therefore, PKCδ and YAP can be utilized as targets to regulate the activity of hepatic stellate cells for the treatment of fibrosis.

## 4. Discussion

In this study, we observed that TPA induced transient G2/M arrest, growth inhibition, and a decrease in the expression of α-SMA in LX-2 cells. In addition, TPA increased the phosphorylation of PKCδ and YAP and decreased the nuclear translocation of YAP. However, administration of the pan-PKC inhibitor Gö 6983 ameliorated these TPA-induced responses in LX-2 cells. Our results suggest that the regulation of the activity of PKCδ and YAP can be utilized as a target strategy for controlling the proliferation and fibrosis of hepatic stellate cells.

The majority of the reports suggest that PKC promotes fibrosis. Inhibition of PKCα, -β, and -λ, or θ is involved in antifibrotic or fibrotic effects, respectively [22,54,55,56]. Here, the phosphorylation of PKCα/β, PKCθ, and PKCζ/λ was gradually decreased with time compared to the control group. Since the antifibrotic effect induced by TPA in LX-2 cells may be regulated by dephosphorylation of PKCα/β and PKCζ/λ, further study is needed to evaluate whether a specific activator of the PKC isoform can modulate LX-2 fibrosis.

PKCδ affects not only cell growth and proliferation but also inflammatory responses and the activation of inflammatory cells [57,58,59,60,61]. In addition, the PKCδ expression has been reported to increase fibrotic tissues, with PKCδ regulating the expression of collagen and α-SMA genes [62,63,64,65]. However, in this study, the expression of α-SMA and COL1A1 was significantly suppressed in LX-2 cells, despite the TPA-induced increase in the expression and phosphorylation of PKCδ. The pan-PKC inhibitor, Gö 6983 decreased the phosphorylation of PKCδ, resulting in the recovery of the growth rate and levels of expression of α-SMA and COL1A1. Consistent with our results, Karhu et al. reported that TPA reduced cell viability and the expression of α-SMA in cardiac fibroblasts via c- and nPKCs [48], despite not showing whether TPA induced the phosphorylation of PKCs. Therefore, PKCs, including PKCδ, are considered important molecules in regulating fibrosis or antifibrosis in different cell types. Recently, the Hippo pathway has been shown to contribute to the pathogenesis of fibrosis, in which hyperactive YAP and TAZ accumulate in both the epithelial and stromal compartments of fibrotic tissues, including the lung, kidney, liver, heart, skin, and tumor [66]. YAP and TAZ are the main downstream effectors of the mammalian Hippo pathway, which exerts a crucial role in controlling tissue and organ development, fibrosis, and tumorigenesis [67]. Gong et al. [68] suggested that TPA-activated PKCs might be linked to the activation of YAP/TAZ. Briefly, they found that TPA-activated PKCs induced the rapid and potent dephosphorylation of YAP (Ser-127) in HEK293A, HeLa, and U251 cells [68]. Conversely, TPA induced the phosphorylation of YAP/TAZ in MEF, A549, and Swiss3T3 cells. They explained that cPKCs promoted the dephosphorylation and activation of YAP, but overexpression of nPKC (δ, θ, or ε) blocked the TPA-induced dephosphorylation of YAP, with nPKC (η) promoting the phosphorylation of YAP in HEK293A cells. To confirm the phosphorylation or dephosphorylation status of YAP, HEK293A, HeLa, and U251 cells or MEF, A549, and Swiss3T3 cells were treated with TPA under serum-free or complete conditions, respectively. In addition, although serum induces the dephosphorylation of YAP in HEK293A cells, Gong et al. showed that the ectopic expression of PKCε induced the phosphorylation of YAP even in the presence of serum. In particular, the ectopic expression of PKCδ, PKCθ, and PKCη induced a mild-to-moderate increase in the phosphorylation of YAP [68]. Here, we used media supplemented with serum to treat LX-2 cells with TPA and then observed that the level of expression and phosphorylation of PKCδ was gradually increased. Therefore, our results suggest that in LX-2 cells in the presence of serum, TPA induces the expression and phosphorylation of PKCδ and the phosphorylation and cytosolic localization of YAP, leading to the suppression of the expression of α-SMA. Thus, we hypothesized that the regulation of YAP phosphorylation in different cells depends on the level of expression and phosphorylation of PKCs and the presence of serum. Indeed, at 24 h after seeding LX-2 cells under serum conditions, the expression of α-SMA (Appendix A) and nuclear localization of YAP (Appendix A) was rarely observed, whereas both the nuclear translocation of YAP and expression of α-SMA at day two was significantly increased compared with those at day zero control cells (Appendix A). These results suggest that the activity of LX-2 cells can be regulated depending on the serum status; more precisely, in the presence of serum, TPA phosphorylates PKCδ, then phosphorylates YAP, and inhibits the nuclear translocation of YAP, resulting in the suppression of the activation of HSCs.

To further demonstrate that PKCδ and YAP are involved in the suppression of the activation of HSCs, additional studies are needed to demonstrate the regulation of PKCs and YAP functions through the overexpression or downregulation of their expression.

## Figures and Tables

**Figure 1 cells-12-00091-f001:**
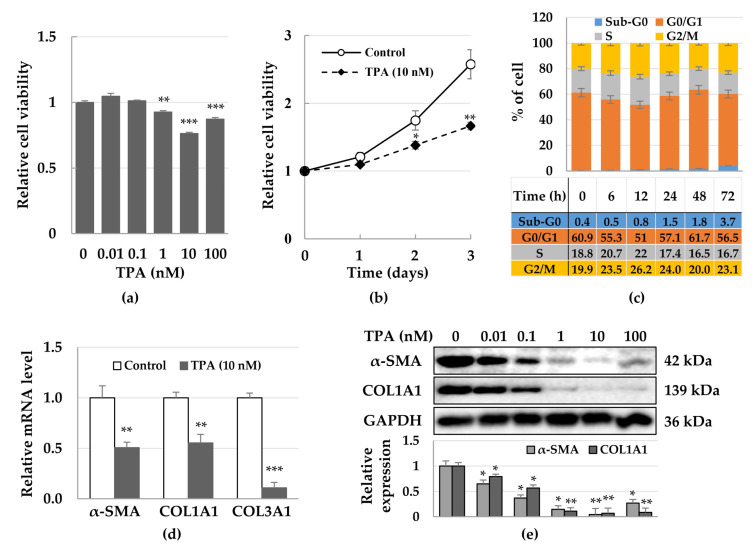
Inhibition of growth and expression of α-SMA in TPA-treated LX-2 cells. LX-2 cells were treated with up to 100 nM TPA for 2 days and then cell viability and expression of α-SMA were analyzed using MTT assay and immunoblotting, respectively. On day 2, mRNA expression was assessed using qPCR, while the cell cycle was examined at the indicated timepoints using flow cytometry. (**a**) LX-2 cell viability according to TPA concentration. Data are presented as the mean ± SD of four independent experiments. ** *p* ≤ 0.01 and *** *p* < 0.001. (**b**) Proliferation of LX-2 cells treated with 10 nM TPA. Data are presented as the mean ± SD of four independent experiments. * *p* ≤ 0.05 and ** *p* ≤ 0.01. (**c**) Cell cycles of TPA-treated LX-2 cells over time. The percentage of cells was analyzed using 2 × 10^4^ cells in indicated timepoints. Data are presented as the mean ± SD of three independent experiments. (**d**) Expression of *α-SMA*, *COL1A1*, and *COL3A1* mRNA in TPA-treated LX-2 cells. All qPCR reactions were performed in triplicates. Expression of *GAPDH* was used for normalization. The 2^−(ΔΔCq)^ method was used to calculate relative fold changes in mRNA expression. (**e**) Expression of α-SMA and COL1A1 in TPA-treated LX-2 cells. Data of (**d**,**e**) are presented as the mean ± SD of three independent experiments. * *p* ≤ 0.05, ** *p* ≤ 0.01, and *** *p* < 0.001.

**Figure 2 cells-12-00091-f002:**
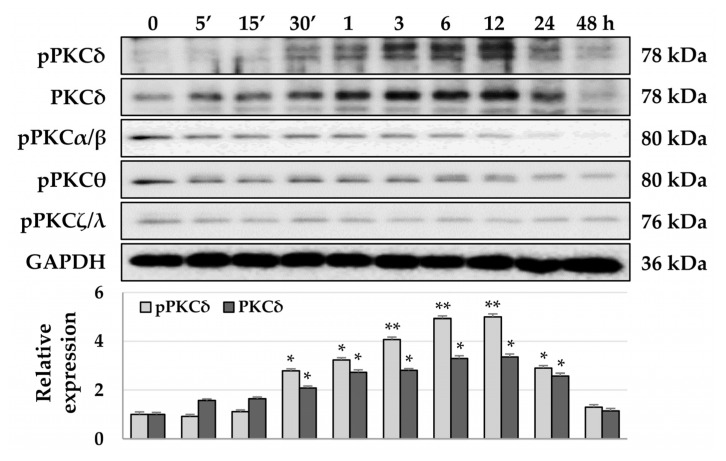
Phosphorylation of PKCs in TPA-treated LX-2 cells. LX-2 cells were treated with 10 nM TPA for the indicated time, and PKC phosphorylation was detected using the phospho-PKC antibody sampler kit (cell signaling technology). As only the phosphorylation of PKCδ was increased in TPA-treated LX-2 cells, we further confirmed the expression of total PKCδ. The intensity of protein expression was quantified using densitometry with Image J and its relative expression was normalized against that of GAPDH. Data are presented as the mean ± SD of three independent experiments. * *p* ≤ 0.05 and ** *p* ≤ 0.01.

**Figure 3 cells-12-00091-f003:**
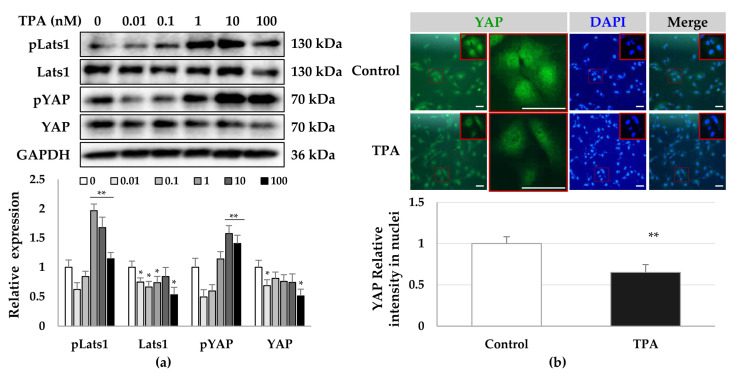
Phosphorylation and cellular distribution of YAP in TPA-treated LX-2 cells. LX-2 cells were treated with TPA (0.01–100 nM) for 24 h. The phosphorylation and cellular distribution of YAP were detected using immunoblotting and immunocytochemistry, respectively, in LX-2 cells treated with 10 nM TPA for 24 h. (**a**) Phosphorylation of LATS1 and YAP in LX-2 cells. The intensity of protein expression was quantified using densitometry with Image J and its relative expression was normalized against that of GAPDH. Data are presented as the mean ± SD of three independent experiments. * *p* ≤ 0.05 and ** *p* ≤ 0.01. (**b**) Cellular distribution of YAP in TPA-treated LX-2 cells. The nuclear fluorescence intensity of YAP was quantified using densitometry with Image J. Nuclear fluorescence intensity was analyzed from three random fields, with over 30 cells counted per field. Data are presented as the mean ± SD of three independent experiments. ** *p* ≤ 0.01. Scale bar, 20 μm.

**Figure 4 cells-12-00091-f004:**
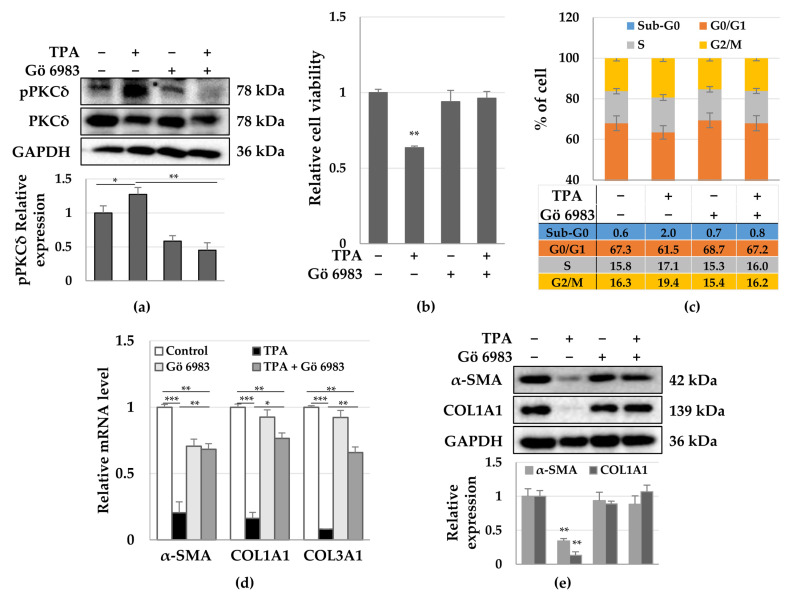
Effects of the pan-PKC inhibitor Gö 6983 on the proliferation and fibrosis of TPA-treated LX-2 cells. LX-2 cells were treated with TPA (10 nM) or Gö 6983 (1 μM) or both, and then cell viability, cell cycle, and expression of α-SMA were analyzed. (**a**) PKCδ phosphorylation in LX-2 cells treated with TPA, Gö 6983, or both for 12 h. The intensity of pPKCδ was quantified using densitometry with Image J and its relative expression was normalized against that of GAPDH. Data are presented as the mean ± SD of three independent experiments. * *p* ≤ 0.05 and ** *p* ≤ 0.01. (**b**) LX-2 cell viability after treatment with TPA or Gö 6983 or both for 48 h. Data are presented as the mean ± SD of four independent experiments. ** *p* ≤ 0.01. (**c**) Cell cycles of LX-2 cells treated with TPA or Gö 6983 or both. The percentage of cells was analyzed using 2 × 10^4^ cells treated with TPA or Gö 6983 or both for 12 h. Data are presented as the mean ± SD of three independent experiments. (**d**) Expression of *α-SMA*, *COL1A1*, and *COL3A1* mRNA in LX-2 cells treated with TPA or Gö 6983 or both for 48 h. All qPCR reactions were performed in triplicates. Expression of *GAPDH* was used for normalization. The 2^−(ΔΔCq)^ method was used to calculate relative fold changes in mRNA expression. Data are presented as the mean ± SD of three independent experiments. * *p* ≤ 0.05, ** *p* ≤ 0.01, and *** *p* < 0.001. (**e**) Expression of α-SMA and COL1A1 in LX-2 cells treated with TPA or Gö 6983 or both for 48 h. * *p* ≤ 0.05, ** *p* ≤ 0.01, and *** *p* < 0.001. The intensity of protein expression was quantified using densitometry with Image J and its relative expression was normalized against that of GAPDH. Data are presented as the mean ± SD of three independent experiments. ** *p* ≤ 0.01.

**Figure 5 cells-12-00091-f005:**
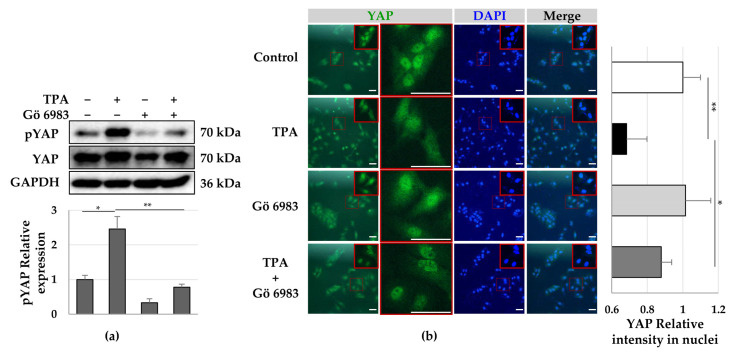
Phosphorylation and cellular distribution of YAP in LX-2 cells treated with TPA or Gö 6983 or both. LX-2 cells were treated with TPA (10 nM) or Gö 6983 (1 μM) or both for 24 h. The phosphorylation and cellular distribution of YAP were detected using immunoblotting and immunocytochemistry, respectively. (**a**) YAP phosphorylation in LX-2 cells. The intensity of pYAP was quantified using densitometry with Image J and its relative expression was normalized against that of GAPDH. Data are presented as the mean ± SD of three independent experiments. * *p* ≤ 0.05 and ** *p* ≤ 0.01. (**b**) Cellular distribution of YAP in LX-2 cells treated with TPA or Gö 6983 or both. The nuclear fluorescence intensity of YAP was quantified using densitometry with Image J. Nuclear fluorescence intensity was analyzed from three random fields, with over 30 cells counted per field. Data are presented as the mean ± SD of three independent experiments. * *p* ≤ 0.05 and ** *p* ≤ 0.01. Scale bar, 20 μm.

**Table 1 cells-12-00091-t001:** List of primer sequences used in this study.

Gene	Forward Primer	Reverse Primer	Size, bp	Accession #
COL1A1	5′-CAGGAGGCACGCGGAGTGTG-3′	5′-GGCAGGGCTCGGGTTTCCAC-3′	263	NM_000088.4
COL3A1	5′-TCCCGGTCCTGCTGGTTCCC-3′	5′-ATGGCAGCGGCTCCAACACC-3′	390	NM_000090.4
α-SMA	5′-GACAATGGCTCTGGGCTCTGTAA-3′	5′-CTGTGCTTCGTCACCCACGTA-3′	149	NM_001613.4
GAPDH	5′-CAAGGCTGAGAACGGGAAGC-3′	5′-AGGGGGCAGAGATGATGACC-3′	194	NM_001256799.3

## Data Availability

The data that support the findings of this study are available from the corresponding author upon reasonable request.

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
