# Peer review of "12-O-tetradecanoylphorbol-13-acetate Reduces Activation of Hepatic Stellate Cells by Inhibiting the Hippo Pathway Transcriptional Coactivator YAP"

_cells, 2022, doi:10.3390/cells12010091_

Round 1

Reviewer 1 Report

The manuscript only analyses a cell line and the effects seem to be transient. The overall advance reported in the manuscript is relatively minor

The manuscript explores the effect of TPA on the fibrotic profile of a hepatic stellate cell line (LX-2). The results show that TPA treatment reduces cell viability, decreases transiently the proportion of cells in G1 stage, increases activated PKCd and Hippo pathway components, LATS1 and YAP, phosphorylation in correlation with a decrease of YAP activated (nuclear). Alternatively, a pan-PKC inhibitor, Gö 6983, reverted the effects of TPA. In general, all of the findings are consistent with a TPA action on HSCells to reduce the fibrotic profile of the cells as shown by decreased expression of fibrosis markers such as aSMA and collagens type 1 and 3. The results are interesting and, accordingly to the authors claim, could offer a potential application to reduce severity of hepatic fibrosis. However, the analysis of the data has severe problems.

1. There is not quantification of the western-blots and immunofluorescences of YAP distribution and, together with the cell cycle analysis, lacks statistical analysis. Moreover, as shown in the original figures, it seems that all these experiments were done only once. The authors have to state the number of repetitions in measurements made for each study and to show quantifications and statistical analysis.

2. Several western-blots show abundant bands of cross reactivity and the same antibody in distinct blots looks totally different, e. g. aSMA in Fig 1 compared to Fig 4 or Supp. Fig 2. Similarly, in the pPKCd blot in Fig. 2 is not possible to identify the specific band and looks different to Fig 4. In Fig 2 blot it seems that there is an increase in total PKCd but is not clear that the phosphorylated fraction increases with the time of TPA treatment. Blocking with skim milk is not recommendable when phospho-specific antibodies are used.

3. TPA has been used as an activator of cPKC and nPKC. What is the explanation that the authors do not observe activation of pPKCa/b? On the contrary, they sustain to observe a decrease, although is very difficult to see when there is not quantification of the bands. Finally, in Fig 2 the blot to pPKCa/b only shows one band.

4. In the abstract and in line 267, the authors use the sentence “TPA phosphorylates PKCd, LATS1 and YAP”. This is confusing and it seems that TPA itself is the protein kinase. Should be used “induces, stimulates, leads to ….”

5. The authors use the term proliferation when in fact are quantifying cell viability. They do not show data of cell death or apoptosis.

Minor

1. In line 79, the authors mention previous experiments but there is not reference

2. In line 173 mention several cell lines but in Mat. Met. lacks information about these cells and culture condition.

3. Fig 4c should be Fig 4a

Author Response

Comments and Suggestions for Authors

We are grateful for the reviewer’s insightful and constructive comments on our manuscript. We have addressed all the comments carefully to improve our manuscript. Our responses to the comments are given below:

The manuscript only analyses a cell line and the effects seem to be transient. The overall advance reported in the manuscript is relatively minor

The manuscript explores the effect of TPA on the fibrotic profile of a hepatic stellate cell line (LX-2). The results show that TPA treatment reduces cell viability, decreases transiently the proportion of cells in G1 stage, increases activated PKCd and Hippo pathway components, LATS1 and YAP, phosphorylation in correlation with a decrease of YAP activated (nuclear). Alternatively, a pan-PKC inhibitor, Gö 6983, reverted the effects of TPA. In general, all of the findings are consistent with a TPA action on HSCells to reduce the fibrotic profile of the cells as shown by decreased expression of fibrosis markers such as aSMA and collagens type 1 and 3. The results are interesting and, accordingly to the authors claim, could offer a potential application to reduce severity of hepatic fibrosis. However, the analysis of the data has severe problems.

  1. There is not quantification of the western-blots and immunofluorescences of YAP distribution and, together with the cell cycle analysis, lacks statistical analysis. Moreover, as shown in the original figures, it seems that all these experiments were done only once. The authors have to state the number of repetitions in measurements made for each study and to show quantifications and statistical analysis.

Response: We greatly appreciate your comment. We have specified the number of repetitions of measurements performed for each study and indicated the quantification and statistical analysis in all figures.

  1. Several western-blots show abundant bands of cross reactivity and the same antibody in distinct blots looks totally different, e. g. aSMA in Fig 1 compared to Fig 4 or Supp. Fig 2. Similarly, in the pPKCd blot in Fig. 2 is not possible to identify the specific band and looks different to Fig 4. In Fig 2 blot it seems that there is an increase in total PKCd but is not clear that the phosphorylated fraction increases with the time of TPA treatment. Blocking with skim milk is not recommendable when phospho-specific antibodies are used.

Response: After detecting one antibody via immunoblotting, we frequently strip the Western blot and reprove it to check the expression of other proteins. The reason why multiple western blots show abundant bands of cross-reactivity and the same antibody looks completely different on separate blots is because the stripping is not perfect in our system. However, our interpretation of the results is not erroneous, as Western blot results can be confirmed by the expected size of the desired protein. Furthermore, although we agree that blocking with skim milk is not recommended when using phosphorus-specific antibodies, our findings did not vary substantially from blocking with 5% FBS in TBST.

  1. TPA has been used as an activator of cPKC and nPKC. What is the explanation that the authors do not observe activation of pPKCa/b? On the contrary, they sustain to observe a decrease, although is very difficult to see when there is not quantification of the bands. Finally, in Fig 2 the blot to pPKCa/b only shows one band.

Response: TPA is used as an activator for cPKC (α, βI, βII, and γ) and nPKC (α, βI, βII, and γ), but phosphorylation of PKCα/β is reduced in LX-2 cells by TPA treatment. Thus, activation of PKCα/β by TPA did not occur in LX-2 cells. cPKC or nPKC may not be activated depending on the cell type. In fact, type 2A phosphatase (PP2A) can regulate the dephosphorylation of PKCα and β by TPA (J. Biol. Chem. 283, 6312-6320; J. Biol. Chem. 277, 5322-5329).

We have added the band quantification results to Figure 2.

The antibody included in Cell signaling technology (CST)'s phospho-PKC antibody sampler kit is pPKCα/βII (Thr638), and CST explained that the size of PKCα and βII are similar at 80 kDa.

  1. In the abstract and in line 267, the authors use the sentence “TPA phosphorylates PKCd, LATS1 and YAP”. This is confusing and it seems that TPA itself is the protein kinase. Should be used “induces, stimulates, leads to ….”

Response: We have corrected that sentence using "induce".

  1. The authors use the term proliferation when in fact are quantifying cell viability. They do not show data of cell death or apoptosis.

Response: Apoptosis or cell death can be confirmed via microscopic observation and can be quantified via annexin-V staining or cell cycle analysis. In the cell cycle analysis of Figure 1c, approximately 1.8% of the population was in Sub-G0 phase, a dead cell population, at 48 h, but in Figure 1a, 10 nM TPA decreased cell viability by approximately 24% compared to the control group. Therefore, the decrease in LX-2 cell viability by 10 nM TPA is not caused by cell death but by the decrease in proliferation caused by transient cell cycle arrest.

Minor

  1. In line 79, the authors mention previous experiments but there is not reference

Response: We have added a reference to that section accordingly.

  1. In line 173 mention several cell lines but in Mat. Met. lacks information about these cells and culture condition.

Response: Culture conditions of the cell lines have been described in Figure S1 legend.

  1. Fig 4c should be Fig 4a

Response: Thank you for the suggestion. We have reordered Fig. 4c into Fig. 4a.

Reviewer 2 Report

Kim and colleagues investigated the antifibrotic effct of protein kinase C (PKC) by using the PKC activator TPA in a cell line. Treatment of cells with TPA decreased the growth rate of LX-2 cells and inhibited the expression of α-SMA and COL1A1. In addition, TPA modulates components of the Hippo pathway. Taken together, the study shows that PKC exerts an antifibrotic effect by inhibiting the Hippo pathway in hepatic stellate cells.

The paper is written well and results are comprehensible. However, there are some major issues that need to be addressed as following before the paper is published.

Introduction

In general, what was the reason to investigate only the PKC isoform? Was an expression analysis of PKC isoforms in cell line performed before the study started?

Page 1, lines 79-81: “In addition, we found […].” Are these unpublished results?

Page 1, line 85:  What was the reason to use LX-2 cells. Short characterization of cells should be included.

Clinical aspect of hepatic stellate cells and their association with liver fibrosis should be more described.

Is there any clinical application known for TPA? Are PKC and YAP clinical contact points to use for therapeutic intervention?

Material and Methods

Page 3, line 105: Is this cell line from human or mouse? Please specify this point.

Include description of HepG2 and Huh-7 cells, because results obtained with these cell lines are shown (Page 4, lines 175-178).

Page 3, Table 1. Please add the size of primer products (bp) and GenBankID to better characterize primer sequences.

Please be correct when describing statistical tests. Students t-test is a parametric test used for normally distributed data. Mann-Whitney is a non-parametric statistical test.

Results

Effect of PKC was investigated after 12 hours, but effects of YAP was first analyzed after 24 h? The reason for this is not comprehensible.

Page 5, line 213: Authors investigated many other PKC isoforms, which were decreased, but this finding was not discussed later.

Page 5, line 190: COL1A1 should be written cursive, because gene is mentioned.

Considering Western Blot data:

- add the molecular weights of proteins

- number of samples (n) of each WB experiment is missing

- quantification of WB data is not shown, but only one representative picture of one sample!?

Considering graphs:

Add the data points to each bar. The error bars appear to be very small, which is uncommon and should be checked, because e.g. SEM normally appears to be smaller.

Considering immunofluorescence data:

number of samples (n) are missing and pictures should be enlarged.

Figure 1b. Replace the „minus sign“ by „control“

Figure 4d. Error bars are missing in the control. Control and TPA+Gö 6983 are not compared. Titel of figure 4 is incomplete, COL1A1 and COL3A1 are missing.

 Discussion

In general, the discussion is very short. Results obtained with other PCK isoforms are not discussed, although a decrease of these PKCs is shown (Figure 2).

Hippo-pathway with components should be more explained and illustrated.

Author Response

Comments and Suggestions for Authors

We are grateful for the reviewer’s insightful and constructive comments on our manuscript. We have addressed all the comments carefully to improve our manuscript. Our responses to the comments are given below:

Kim and colleagues investigated the antifibrotic effct of protein kinase C (PKC) by using the PKC activator TPA in a cell line. Treatment of cells with TPA decreased the growth rate of LX-2 cells and inhibited the expression of α-SMA and COL1A1. In addition, TPA modulates components of the Hippo pathway. Taken together, the study shows that PKCẟ exerts an antifibrotic effect by inhibiting the Hippo pathway in hepatic stellate cells.

The paper is written well and results are comprehensible. However, there are some major issues that need to be addressed as following before the paper is published.

Introduction

In general, what was the reason to investigate only the PKCẟ isoform? Was an expression analysis of PKC isoforms in cell line performed before the study started?

Response: As shown in Figure 2, only PKCẟ was phosphorylated by TPA in LX-2 cells and its expression increased, therefore, we focused on PKCẟ. We understand why the reviewer asked this question. Therefore, discussions on dephosphorylated PKCs have been added to the Discussion section.

Page 1, lines 79-81: “In addition, we found […].” Are these unpublished results?

Response: We have added the relevant references to that section.

Page 1, line 85:  What was the reason to use LX-2 cells. Short characterization of cells should be included.

Response: We have added the reason for using LX-2 cells in section 3.1 as follows.

In addition, since LX-2 cells with low basal expression of α-SMA are better suited for the TGF-β-induced hepatic stellate cell activation assay and the expression of α-SMA was reduced by TPA treatment compared with the control group, LX-2 cells were used for this experiment (Figure S2).

Clinical aspect of hepatic stellate cells and their association with liver fibrosis should be more described.

Response: As described in the introduction, I believe that the relationship between the clinical aspect of hepatic stellate cells and liver fibrosis cannot be explained without mentioning that hepatic stellate cells play an important role in the progression of liver fibrosis. In fact, TGF-β is known to be an important factor inducing the activity of hepatic stellate cells, but clinical studies on mitigating liver fibrosis with TGF-β receptor inhibitors have not been undertaken. Since clinical studies targeting hepatic stellate cells are predicted to have many side effects, it seems difficult to explain the association between the clinical features of hepatic stellate cells and liver fibrosis.

Is there any clinical application known for TPA? Are PKCẟ and YAP clinical contact points to use for therapeutic intervention?

Response: Currently, nine clinical studies using TPA are registered on https://www.clinicaltrials.gov, all of which are in the field of tumor treatment, and there are no clinical studies on fibrosis relief. Since TPA can act as a tumor promoter, it is considered possible to study fibrosis control using PKC activators that do not affect tumor growth. Therefore, in the future, we plan to use PKC activator PEP 005, BRY1, or PDBu and YAP inhibitor verteporfin to study whether clinical treatment is possible via the regulation of PKCẟ and YAP activity.

Material and Methods

Page 3, line 105: Is this cell line from human or mouse? Please specify this point.

Response: LX-2 cells were specified as being of human origin.

Include description of HepG2 and Huh-7 cells, because results obtained with these cell lines are shown (Page 4, lines 175-178).

Response: This information was already described in Figure S1 legend.

Page 3, Table 1. Please add the size of primer products (bp) and GenBankID to better characterize primer sequences.

Response: We have added the size (bp) and accession number of the primer products in Table 1.

Please be correct when describing statistical tests. Students t-test is a parametric test used for normally distributed data. Mann-Whitney is a non-parametric statistical test.

Response: Thank you for the comment. We have modified the description of statistical tests accordingly (Section 2.8).

Results

Effect of PKCẟ was investigated after 12 hours, but effects of YAP was first analyzed after 24 h? The reason for this is not comprehensible.

Response: Gö 6983 inhibited the phosphorylation of YAP by TPA at 24 h (Figure 5). In addition, although not discussed in this study, in the subcellular fractionation experiment, TPA did not affect the nuclear translocation of YAP at 12 h, but at 24 h, and the level of YAP in nuclei was decreased, as shown in figure below (unpublished data). Therefore, we considered 24 h as the optimal time to check the phosphorylation and subcellular localization of YAP.

Page 5, line 213: Authors investigated many other PKC isoforms, which were decreased, but this finding was not discussed later.

Response: Thank you for pointing this out. A discussion of the dephosphorylation of the PKC isoforms (PKCα/β, PKCθ, and PKCζ/λ) has been added to Discussion section (Lines 302–307).

Page 5, line 190: COL1A1 should be written cursive, because gene is mentioned.

Response: Thank you for this comment. We have corrected it

Considering Western Blot data:

- add the molecular weights of proteins

Response: We have added the molecular weights of the proteins.

- number of samples (n) of each WB experiment is missing

Response: We have indicated the number of WB replicates in the figure legend.

- quantification of WB data is not shown, but only one representative picture of one sample!?

Response: We have added the quantification of WB data.

Considering graphs:

Add the data points to each bar. The error bars appear to be very small, which is uncommon and should be checked, because e.g. SEM normally appears to be smaller.

Response: We fully agree with the reviewer's comment. However, we think the current graphs are the best for readers to easily understand our results.

Considering immunofluorescence data:

number of samples (n) are missing and pictures should be enlarged.

Response: We have added the number of samples (n) to the figure legends. The original figures already include enlarged pictures, and we have shown the YAP-stained area slightly larger.

We also have enlarged the picture a bit more and added a graph showing the YAP relative intensity in the nucleus as we believe it would be more helpful in understanding the picture than further enlarging the image.

Figure 1b. Replace the „minus sign“ by „control“

Response: We have replaced the minus sign by “control”.

Figure 4d. Error bars are missing in the control. Control and TPA+Gö 6983 are not compared. Titel of figure 4 is incomplete, COL1A1 and COL3A1 are missing.

Response: The reviewer's comments are now reflected in Fig. 4d and the title of Figure 4 has been modified.

 Discussion

In general, the discussion is very short. Results obtained with other PCK isoforms are not discussed, although a decrease of these PKCs is shown (Figure 2).

Response: A discussion of the dephosphorylation of the PKC isoforms (PKCα/β, PKCθ, and PKCζ/λ) has been added to Discussion section (Lines 302–307).

Hippo-pathway with components should be more explained and illustrated.

Response: Thank you for the suggestion. Details regarding the components of the hippo pathway have been added to Discussion section (Lines 319–324).

Round 2

Reviewer 1 Report

The authors addressed all the questions of this referee

Reviewer 2 Report

The authors have addressed most issues and substantially improved the manuscript.